# Perfective *-le* Use and Consciousness-Raising among Beginner-Level Chinese Learners

Yi Xu 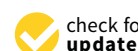

Department of East Asian Languages and Literatures, University of Pittsburgh, Pittsburgh, PA 15260, USA;
xuyi@pitt.edu

**Abstract:** Within the framework of explicit learning and consciousness-raising, this study investigates patterns in the use of *-le* in authentic classroom tasks by beginner-level learners of Chinese as a foreign language (CFL). It also explores the role and the processes of student-centered consciousness-raising in explicit knowledge building. Twenty-five participants completed a grammaticality judgment task, an interactive role-play task, and a written editing task. The experiment group received role-play sheets with explicit forms of *-le* provided, and participants engaged in rule induction of *-le* in forbidden context in the role-play session. Results showed that beginner-level learners' difficulty with *-le* use manifested in different ways in these tasks, and *-le* underuse occurred more than overuse in the control group's oral role-play task. Consciousness-raising through unguided small group rule induction supported participants' learning of *-le* usage constraints, shown by differences between the control and experiment groups' performances in the posttest. Through a qualitative analysis of participants' analytical talk transcripts, the processes and outcomes of small group rule induction are examined and discussed.

**Keywords:** Chinese as a foreign language; consciousness-raising; obligatory and forbidden contexts; perfective aspect marker; rule induction

## 1. Introduction

Perfective aspect marker *-le* use in Chinese is an important and challenging grammatical area for Chinese as a foreign language (CFL) learners. The topic has drawn interest from linguists and language acquisition researchers since decades ago (e.g., Wen 1995; Wen 1997). More recently, some attempts have been made in investigating effective pedagogical interventions within the framework of form-focused consciousness-raising (Yuan 2012; Yuan 2019). Meanwhile, there are very few empirical studies that have observed learners' performance in regular language classrooms. Pedagogical proposals that integrate form-focused instruction with communicative activities are also lacking. The present research investigates beginner-level CFL learners' use of the perfective aspect in classroom role-play tasks, and discusses the effect of consciousness-raising and rule induction in learners' acquisition of *-le*.

## 2. Literature Review

### 2.1. L2 Acquisition Studies of Aspect Marker -le

Particle *-le* as a perfective aspect marker is a crucial component of the Chinese language structure expressing viewpoints. Beginner-level CFL learners are generally introduced to both the perfective marker *-le* and sentence-final *-le*. The two have distinct usages, though they may also incidentally overlap (Xiao and McEnery 2004). Unless otherwise specified, *-le* in this study refers to the former. Studies on *-le* point to a few common usages of the aspect marker, which are directly associated with certain forms. For instance, Li and Thompson (1989) pointed out that *-le* is used with bounded events

and an event can be bounded in one of four ways: (1) by being a quantified event; (2) by being a definite or specific event; (3) when the verb is inherently bounded; or (4) when being the first event in a sequence (pp. 185–86). Such "boundedness" features are respectively reflected in the following forms: (1) V-*le*- Numeral-Measure Word-(O) (referred to as V-*le*-NM below), (2) V-*le*-definite or specific NP, (3) telic and resultative V-*le*, including resultative verb compound (RVC)-*le*, and (4) V1-*le*(O)-V2[1]. In acquisition studies, some earlier researchers summarized common patterns of -*le* usage in similar ways (Ke 2005; Yuan 2012). To some extent, textbooks also introduce the form-function mapping as grammatical rules or language patterns to learners.

Correspondingly, there are a list of contexts where -*le* is disallowed. For instance, habitual and state verbs are generally incompatible with perfective aspect across different languages (Comrie 1976; Smith 1991). -*le* also tends not to go well with "say" types of verbs indicating direct or indirect speech, or verbs taking a clausal object (Duff and Li 2002, pp. 424–25; T'ung and Pollard 1982). Because these negative contexts can be categorized in different ways, and not all of them are suitable pedagogical material for beginner-level CFL learners, the present study deals specifically with the three types of constraints in -*le* usage summarized in Table 1. That is, -*le* is generally not used if the event was a habitual activity, a stative event, or expressions of direct or indirect speech, even if such events took place in the past.[2] These constraints can thus serve as useful pedagogical examples illustrating differences between the English past tense and the Chinese perfective aspect. The constraints are referred to as -*le* forbidden contexts in this paper.

**Table 1.** Forbidden context of -*le* and examples.

| -*le* Forbidden Environment | Sample Sentences |
| --- | --- |
| Habitual activity | *Wo qunian changchang youyong (\*le)*. "I often swam last year." |
| State verbs | *Zuotian wo juede (\*le) bu-shufu*. "Yesterday I felt uncomfortable." |
| Expressions of direct/indirect speech | *Wo gaosu (\*le) ta, wo jintian bu qu xuexiao*. "I told him that I would not go to school today." |

Previous studies also referred to some other -*le* forbidden contexts that may need further scrutiny. Duff and Li (2002, p. 425), citing T'ung and Pollard (1982), claimed that verbs taking a clausal object also do not go with perfective -*le* and gave the following example where -*le* insertion would be ungrammatical: *zuotian women jueding (\*le) qu kan na ge dianying* ("Yesterday we decided to go see that movie"). However, the examples given to support this purported "-*le* forbidden rule" tend to involve psych verbs, other state verbs, or verbs related to indirect speech (such as "plan", "decide", "claim"). Thus, it remains dubious to what extent the "clausal object" rule is effective beyond what has already been covered by in stipulations in Table 1. -*le* may also have other constraints. For instance, it typically does not occur in negated form (*wo mei chifan*, "I did not eat"), but as this constraint has more to do with syntactic behaviors rather than semantic aspect, it is of a rather different nature.

In second language acquisition (SLA) studies of Chinese, empirical evidence indicates that -*le* emerges early in learners' language acquisition. In the seminal work of Wen (1995), the author reported that beginner-level learners who had studied Chinese for 14 months used -*le* in several forms, including, V1-*le*(O)-V2, V-NM, V-telic verbs, with 79%, 74%, and 78% accuracy rates, respectively. Similarly high accuracy rates were reported in Wen (1997) among learners with 15–27 months' experience learning

---

[1]  V, O, and NP respectively refer to "verbs", "objects", and "noun phrases". Many but not all RVCs obligatorily require -*le*. According to Jin and Hendriks (2005, p. 71), result-state RVCs with a state verb complement "do not [ . . . ] depict a process with inherent endpoint" and instead "present the result from a process". In these result-state RVCs (*xi-ganjing* "wash-clean"; *xiuli-hao* "fix well"), -*le* is needed to indicate completion of the event. See *Note 6* for examples of RVCs that can optionally take -*le*.

[2]  An anonymous reviewer pointed out that -*le* may be used in some situations of indirect speech, such as in correcting a wrong assumption. There can be marked situations where -*le* is possible in expressing (in)direct speech for emphasis or contrast. For instance, *laoshi yijing shuo-le mingtian bu shangke, ni weishenme hai yao qu xuexiao?* "The teacher already said that there is no class tomorrow, (so) why would you still go to campus?" Care was taken so that materials in this study did not involve these contrasting situations where -*le* may be warranted.

Chinese. Meanwhile, *-le* is infamously difficult even for advanced-level learners (e.g., Yang et al. 1999). One of the major causes of challenge is negative L1 transfer (Tong and Shirai 2016; Wen 1997). Learners with an L1 background of English were found to have a tendency to use *-le* as a past tense marker. As shown by participants' self-reflection reported in Duff and Li (2002), learners may oversupply *-le* in written production tasks, even if they were explicitly aware that the two were not the same. Tong and Shirai (2016) hypothesized that their beginner- to intermediate-level participants were not sensitive enough to the lexical aspect of the verbs. Wen (1997) observed that her participants often adopted a meaning-based approach and relied on contextual cues, such as time adverbials or other expressions (*yiqian* "in the past"; . . . *de shihou* "at the time of"). While such cues can sometimes be helpful, they are not reliable indicators of *-le* usage, as they are more indicators of past time frame than indicators of viewpoint or boundedness. Researchers also note that the optionality of aspect marking in some contexts in Chinese constitute another source of difficulty for learners. Chen and Shirai (2010) observed that Chinese children erroneously used perfective *-le* with state verbs in early stages of their L1 acquisition, and those authors hypothesized that such deviations from standard usages are due to *-le* use in optional context, making it difficult for learners to induce rules. Corresponding to this finding in L1 acquisition, L2 corpus studies reported similar results. Yang et al. (1999), using corpus of Chinese learners' composition, observed that learners tend to overuse *-le* with stative predicates. In the most recent corpus study examining beginner- to advanced-level L2 learners' speech, Xu et al. (2019) also claimed that across different proficiency levels, learner errors primarily involved ungrammatical use of *-le* with state verbs. These studies argued that overuse was a serious issue in L2 learners' *-le* usage, and attributed it to learners' inadequate knowledge regarding constraints of *-le*.

Researchers have noted that learners' use of *-le* may be subject to task and register variations. For instance, in Duff and Li (2002), the researchers compared learner productions with native speakers' performance. They found that learners underused *-le* in oral narrative tasks, but oversupplied *-le* in a written editing task. Table 2 summarizes the tasks used in previous L2 studies of *-le*.

**Table 2.** Tasks used in previous L2 studies of perfective *-le* acquisition.

| Task Type | Details | Study |
|---|---|---|
| Conversation | Questions designed to elicit target aspect markers. | Wen (1995); Wen (1997) |
| Narrative oral task | Based on a silent film | Duff and Li (2002) |
| | Personal narrative | Duff and Li (2002) |
| | Based on sets of picture sequences | Jin and Hendriks (2005) |
| Picture-based non-narrative oral task | Questions and answers | Wen (1997) |
| | Description of isolated pictures | |
| Picture-based writing task | Narrative | Yuan (2019) |
| | Non-narrative description | Wen (1997) |
| Judgment and written editing | Edit sentences or paragraph-length texts by supplying aspect markers | Duff and Li (2002); Shi (2013); Tong and Shirai (2016); Yang et al. (2000) Yuan (2012) |
| Learner corpus studies | Usage and error analysis from corpora data | Yang et al. (1999) (written) Xu et al. (2019) (speech) |

These earlier studies, using a variety of tasks, have shed light on features of learner language and its development. However, most of them were conducted using experimental tasks or corpus data, and participants' performance might not directly reflect situations in language classrooms. One pedagogical approach attracting and favored by many practitioners is task-based language teaching (TBLT). Meanwhile, there is evidence that classroom-based oral interactive tasks can yield useful production data for SLA research (e.g., Bygate 2001; Kim 2013). The above review suggests that our

knowledge about students' use of *-le* in language classroom tasks is still very limited, and there exists a disconnection between L2 studies of *-le* in experimental environment and real teaching practice.

## 2.2. Language Tasks and Consciousness-Raising

Tasks can be used alongside focus-on-forms instruction. That is, in a focused task design, a particular language form can be purposefully embedded in task design to induce learners to use them while performing the task (Ellis 2001). Communicative tasks can also be used alongside other form-focused approaches, such as consciousness-raising (Ellis 2003). Broadly speaking, consciousness-raising is activities or efforts made to make specific language features salient to learners, and can include techniques such as textual enhancement (e.g., bolding, underlining) so that learners' attention can be drawn to certain linguistic features (Leow 2015, p. 167). More often, consciousness-raising tasks are activities in which learners "perform some operations on or with" L2 data, with the goal of achieving an explicit understanding of specific target language properties (Ellis 1997, p. 160). Learners are asked to discover and verbalize target language rules through discussions. Such "metatalks" or "languaging", Swain (2000, 2006), can be operationalized in different ways. Previous L2 studies in other languages have used instructor-guided rule induction (e.g., Adair-Hauck et al. 2010), group interaction with optional intervening from the instructor (e.g., Smart 2014), and individual rule induction in online context (Cerezo et al. 2016). These studies suggested that consciousness-raising rule induction helps promote learners' explicit knowledge of complex linguistic structure.

Given the complexity of rules associated with *-le* usage and learners' continuous difficulties in using it in target-like ways, a form-focused approach to facilitate the L2 acquisition of *-le* is fitting. However, empirical studies that evaluate the effectiveness of pedagogical treatment of *-le* remain scant. Yuan (2012) is the only study that has specifically examined the role of consciousness-raising, form-focused instruction on learners' development of explicit knowledge regarding *-le.* In Yuan (2012) study, ten and eight participants took part in the experimental and control group, respectively, and completed pretest, posttest, and a delayed posttest in the same written paragraph editing task. The experiment group experienced three one-hour consciousness-raising sessions in which participants discussed, reviewed, and did form-function mapping exercises on several categories of *-le* usages. Significant improvement in learners' performances was reported between the pretest and the two posttests for the experiment group, and significant differences were also found between the control and the experiment group in the delayed posttest in overall *-le* use accuracy and several sub-categories of *-le* structures. Yuan's (2012) study confirmed the effectiveness of consciousness-raising activities in CFL learners' *-le* acquisition. However, because her control group did not spend an equal amount of time engaging in comparable learning activities, the comparative effectiveness of consciousness-raising versus other pedagogical approaches is yet to be determined.

In sum, despite the number of attempts made to investigate the L2 acquisition of *-le*, there remains a gap between finely controlled research and the realities of classroom practice. More research is needed on the effect of consciousness-raising as a form-focused pedagogical intervention in today's communicative, student-centered classroom environment. The present study addresses these gaps. The purpose of the study is twofold. First, it investigates what errors tend to occur in beginner-level CFL learners' use of *-le* in their oral and written performances in language classrooms. Second, it explores the role of consciousness-raising by using role-play sheets with explicit markings of language forms and by having learners engage in small group rule induction. As previous studies pointed to learners' lack of knowledge regarding constraints of *-le* usage (Xu et al. 2019; Yang et al. 2000), promoting learners' perception and understandings of *-le* forbidden context is especially important. This study also intends to investigate rule induction within student groups without intervention from instructors. The following are the study's specific research questions:

First, what types of errors can be observed in beginner learners' use of *-le* in classroom activities, including written tasks and oral role-plays?

Second, in communicative language classrooms, does consciousness-raising lead to higher learning outcomes than regular role-play tasks in promoting the learning of *-le* constraints?

Third, how effective are learner-centered rule induction sessions without instructor mediation?

## 3. Materials and Methods

### 3.1. Participants

Twenty-five CFL students in a Northeastern American university participated in the study. All participants were native speakers of American English. All participants received a rigorous placement test upon their entrance into the Chinese language program. When the data were collected, participants had had eight months of Chinese learning, and had received approximately 200 hours of classroom instruction. Participants were randomly divided into two groups, with 12 participants in the control group (Group C) and 13 in the consciousness-raising experiment group (Group E). In the role-play and rule induction sessions, participants worked in pairs or small groups of three.

Prior to the study, participants had received instructions on several obligatory contexts of *-le* in their regular course curriculum. Their textbook introduced aspect marker *-le* as a particle indicating "the occurrence or completion of an action or event" (Liu et al. 2008, p. 137). The examples that the textbook gave included patterns of V-*le*-NM, V1-*le*(O)-V2, and V-*le*-proper noun. These illustrations and examples correspond to the "bounded event" environment of obligatory *-le* use specified in Li and Thompson (1989). The textbook also pointed out that in negation of past events, *mei* instead of *mei . . . le* is used. All participants had the same lecture class instructor in their first and second semester study. The instructor confirmed that participants were taught these patterns and usages, and that forbidden contexts other than *mei* negation had not been explicitly taught.

### 3.2. Materials

The assessment and treatment materials included: a grammaticality judgment pretest, two versions of role-play sheets, one each for Group C and Group E, respectively, and a written passage editing task sheet similar to the one used in Duff and Li (2002).

The pretest of grammaticality judgement included 16 items, 4 of which were fillers. For the remaining 12 items, 6 were grammatical sentences. Three of the six contained obligatory *-le* and the other three were in negative *-le* context (state verb, habitual past activity, and expressions of direct or indirect speech); the other six were ungrammatical sentences, with *-le* missing in *-le* obligatory contexts, or *-le* inserted in the three *-le* forbidden contexts. The 16 items were presented in a randomized order. Participants were instructed to mark the sentence with a "C" ("correct") if they considered the sentence grammatical, and mark it with an "I" ("incorrect") if they considered it ungrammatical. They were further instructed to correct the errors in ungrammatical sentences. Whereas earlier studies tapping into representations in learner language often used grammaticality judgment, the additional step of asking participants to correct errors is important. As Gass and Mackey (2012) pointed out, a caveat of the traditional grammaticality judgment task is that learner language grammar may be "non-native-like in many ways," and "it is necessary to ask learners to correct any sentences that they judge to be unacceptable" (p. 98). Appendix A presents the grammaticality judgment task stimuli.

The role-play sheet described, in English, six scenarios in which *-le* should either be used or is forbidden. In each scenario, speaker A and B received different instructions regarding the situation and how they should act out. The role-play sheets differed minimally between the two versions. The Group E role-play sheet contained explicit written prompts where the perfective marker should or should not be used. For instance, in one scenario, speaker B would tell speaker A about a series of unfortunate incidents, including feeling unwell, taking the wrong bus, and having a stomach problem after eating bad food. In addition to the English prompt, the role-play sheet gave participants keywords with their correct usages of *-le*, i.e., 觉得 (*juede*)∅ "feel", 坐错了 (*zuocuo-le*) "took the wrong (bus)", 吃坏了(*chihuai-le*) "caused a (stomach) problem due to eating bad food", providing the "Verb-*le*" form

when *-le* was obligatory, and the "Verb-∅" form when *-le* was ungrammatical. The scenarios and event verbs included both obligatory *-le* contexts and several examples of *-le* forbidden contexts in each of the following: (1) with state verbs, (2) with habitual activities, and (3) expressions of (in)direct speech in the past time frame. The role-play sheet provided a total of 16 explicit Verb-∅ forms and 9 Verb-*le* forms.[3] For Group E, a rule induction responses sheet was appended to the role-play sheet. Participants were asked to discuss with their partners, in either Chinese or English, to induce when *-le* should/should not be used in past events, and write down their conclusions. Appendix B provides the instructions and sample scenarios in the role-play sheet for Group E.

The role-play sheet for Group C was the same for all descriptions of the scenario, but differed in that it did not explicitly mark when *-le* should be used. Instead, this version of the role-play sheet simply provided participants with the key verbs (e.g., *juede* "feel", *zuocuo* "take.wrong(.bus)", *chihuai* "eat.bad"). No rule induction response sheet was provided to Group C.

For posttest, a written passage editing task sheet was designed. The task sheet contained a passage with two paragraphs, telling a story of a person's trip to and experience in China all in the past time frame. The passage contained 24 blanks and participants were instructed to provide *-le* where it was needed, and write "/" when it should be absent. Among the 24 blanks, 12 were cases where ungrammatical *-le* occurred in forbidden contexts, with four sentences each in the environment of a state verb, habitual activity, or a speech verb. The remaining 12 blanks were cases where *-le* was obligatory, with a telic verb or RVC, or followed by an object that was definite or quantified. See Appendix C for the written editing posttest task.

Only words that participants had previously learned from textbooks were used in the materials. For the grammaticality judgment and written passage editing task sheets, *pinyin* were provided on top of each sentence. All scenarios described in the role-play sheet were appropriate for participants' proficiency level in terms of vocabulary needed and structure usage. The participants' instructor was consulted in finalization of the material.

*3.3. Procedure*

Both Group E and Group C completed the study on the same day, each in a 50-minute class session in a regular classroom. That is, the two groups completed the study in two different class periods. The researcher, who was incidentally a language instructor herself, monitored classes. Because the study aimed to investigate the effect of consciousness-raising when learner agency is maximized, the researcher instructed participants to complete each step, recorded time, offered assistance with audio-recording, and collected response sheets. She did not offer pedagogical instructions or guidance on completing tasks.

Both groups first completed the grammaticality judgment pretest in a paper-and-pencil format. The procedure took 15 min. Immediately after the pretest sheets were collected, participants were asked to carry out interactive role-play tasks in Chinese with a randomly assigned partner or two partners. Group E and Group C were given their respective role-play sheets.

For Group E, participants used approximately 12 min to act out the scenarios in the role-play sheet in pairs/groups. After they had all completed the role-play tasks, they were instructed to engage in rule induction within their small group by paying attention to usages in the role-play. They were asked to write down their conclusions on a separate sheet and turn in their responses at the end of their discussion. The rule induction discussion took approximately eight minutes. Each pair/group's role-play and rule induction sessions were audio-recorded.

---

[3] This does not mean that participants were necessarily expected to produce all or only these verb forms. As participants often make spontaneous conversations in role-play tasks, these verb forms with and without aspect markers served as examples. In some cases, a specific "Verb-∅" may also be produced by both speakers in a role-play as questions and answers.

For Group C, participants spent approximately 20 min for the role-play task. They were asked to continue studying the role-play situations or switch roles to play again if they completed all scenarios in less than 20 min. Participants in Group C did not engage in rule induction. Their oral production in role-play was audio-recorded.

In the last 15 min of the class, after role-play sheets (and rule induction responses for Group E) were collected, participants were given the written passage editing task as a posttest.

### 3.4. Scoring and Coding

For the grammaticality judgment task, a score of one was assigned if (1) a participant judged a grammatical sentence to be correct or (2) a participant judged an ungrammatical sentence to be incorrect and also made the right corrections. If a participant judged an ungrammatical sentence to be incorrect but did not identify the location of the ungrammaticality or did not offer target-like corrections, the response received zero score. Zero score was assigned in all other situations.

Audio-recordings of participants' role-play were transcribed. For Group C's oral production, grammatical use and errors of *-le* were coded, including overuse, underuse, and other errors. The researcher and a research assistant coded the usages independently, with 92.5% agreement. Discrepancies were resolved through discussion. For Group E, because correct forms of *-le* use were provided explicitly on the role-play sheet, participants almost always had target-like performance in using *-le*. After confirming that was the case, that part of the data did not enter into further analysis.

For the written passage editing task, each blank was scored either one or zero, based on whether the response was target-like or not.

To answer research question one, namely participants' error patterns in using *-le*, we reported participants' performance in the three tasks. For the grammaticality judgment task, all 25 participants' performance was examined. A paired sample *t*-test revealed no significant differences between the two groups' performance in this task (Group C: Mean = 6.1, S.D. = 1.4; Group E: Mean = 5.3, S.D. = 1.7; $t(23) = 1.25$, $p = 0.22$). For participants' performance in the oral role-play task and the written editing task, only Group C participants' performances were considered relevant.

To answer research question two, namely the outcome of consciousness-raising versus regular role-play tasks, comparisons were made between Group E and Group C participants' scores in *-le* forbidden and *-le* obligatory contexts in the written editing posttest.

To answer research question three, namely the outcome and the processes of rule induction within pairs/small groups, each small group/pair's written responses and transcripts of their rule induction sessions were analyzed qualitatively. Coding categories used in previous studies of consciousness-raising rule induction were used. These codes included *labeling* (observations and naming specific forms and tokens), *categorizing* ("commenting on properties shared by several tokens"), *patterning* ("commenting on links between two categories of forms [ . . . ] or between form and meaning", and *rule formulation* (generalizing patterns) (Toth et al. 2013; Cerezo et al. 2016, p. 270). In Cerezo et al.'s (2016) study in which they triangulated learning processes with learning outcomes, they focused on depth of L2 awareness and referred to codes including *noticing and reporting* (attention to specific forms and features), *hypothesis formulation* (which overlaps with the definition of *patterning* above), *rule formulation*, *prior knowledge or experience*[4], and *metacognition* (describing feelings about one's progress). As rule induction in this study was unique in several aspects, the researcher considered other types of learning processes to be possible. After going over the overall transcribed data several times, the researcher developed two more codes, *contrasting* and *rationalizing*, explained in the "Results" section. Following a peer review procedure (Merriam 2009), a colleague was asked to read the transcription and assess if these categories of codes effectively captured the data. Due to the exploratory nature

---

[4]  While earlier studies used the term *activation of prior knowledge* (Tomlin and Villa 1994), *prior knowledge* and experience is used here to inclusively refer to activation of existing linguistic knowledge and recalling experiences of language use.

of the research question, we were not interested in drawing statistical correlations between specific processes and outcome in this study. Instead, we aimed to report different types of cognitive processes manifested in student-centered, interactive rule induction, with qualitative analysis of how they may or may not contribute to explicit learning.

## 4. Results

### 4.1. Learner Performance across Tasks

To examine learner performances in different tasks, first, all 25 participants' performances in the grammaticality judgment task were analyzed. Participants' scores in judging ungrammatical and grammatical prompts were examined separately, in the *-le* obligatory context and *-le* forbidden context, respectively. Table 3 below shows the descriptive statistics of participants' scores in those situations.

**Table 3.** Participants' scores in grammaticality judgment.

|  | *-le* **Obligatory Context** | | *-le* **Forbidden Context** | |
| :---: | :---: | :---: | :---: | :---: |
|  | Grammatical prompt | Ungrammatical prompt | Grammatical prompt | Ungrammatical prompt |
| **Mean (S.D.)** | 2.24 (0.78) | 0.44 (0.58) | 2.20 (0.76) | 0.80 (0.96) |

Participants achieved mean scores of 2.24 and 2.20 out of a maximum score of 3 in the *-le* obligatory context and *-le* forbidden context, respectively. In both contexts, their rate of accuracy in identifying the location of error and correcting it was low (0.44 and 0.80, out of a maximum score of 3). In all but three of the zero-scored responses for ungrammatical prompts, participants were actually not able to identify the location of the error.

Stimuli receiving the lowest scores included an item describing habitual activity and an item involving indirect speech, with one and two participants providing target-like responses only. Other items receiving low scores included sentences where *-le* was missing in an obligatory context, with an example in (1).

(1)

| zuotian | wo | he | pengyou yiqi | kan | *(le) | Harry Potter |
| yesterday | I | and | Friend together | read | LE | Harry Potter |

"I watched Harry Potter together with friend yesterday."

In general, the pretest results indicate that participants often could not identify situations where *-le* deletion or *-le* insertion were needed. For instance, prompt (1) was directly modified from a sample sentence from the participants' textbook (Liu et al. 2008, p. 139), yet the majority of the participants (18 out of 25) could not identify the location of the error. Participants may not have internalized the knowledge regarding usages and constraints of *-le*.

Second, in the interactive role-play task, both *-le* oversupply and undersupply were witnessed. There were 19 tokens of *-le* undersupply in obligatory context, with nine tokens in RVC context. Except for one role-play small group who correctly produced two RVC-*le* forms, participants in all three other role-play groups constantly failed to supply *-le* in RVC contexts. (2)–(4) provide examples. In addition, there were nine missing *-le* in the V-*le*-NM environment (e.g., *zuotian wo chi *(le) ershi zhi jiaozi* "I had 20 dumplings yesterday"), and one missing *-le* in a specific, completed past event (*Women dou qu *(le)* "We all went").

(2)

| Duibuqi | wo zuo-cuo | *(le) | che |
| sorry | I sit-wrong | LE | bus |

"Sorry, I took the wrong bus."

(3)

| Wo | chi-huai | *(le) | duzi |
|---|---|---|---|
| I | eat-bad | LE | stomach |

"I ate wrong food and had a stomach-ache."

(4)

| yinwei | sushe | hen | chao, suoyi | wo | ban-qu | *(le) | gongyu |
|---|---|---|---|---|---|---|---|
| because | dormitory | very | noisy, so | I | move to | LE | apartment |

"Because my dormitory was very noisy, I moved out to an apartment."

In several cases, participants' supply of *-le* seemed arbitrary. They supplied *-le* in one clause containing V-*le*-NM forms, but failed to do so in the conjoined clause with the same structure. (5) provides an example.

(5)

| Wo zuotian | chi-le | shi | ge | shuijiao, | hai | wo | he | *(le) | san | bei | kafei |
|---|---|---|---|---|---|---|---|---|---|---|---|
| I yesterday | eat-LE | ten | CL | dumplings | also | I | drink | LE | three | cup | coffee |

"Yesterday I ate ten dumplings; I also drank three cups of coffee."

Ten tokens of *-le* oversupply in forbidden contexts were found, including four tokens in habitual activities in past time frame, with an example shown in (6), two tokens of ungrammatical *-le* with state verbs, with an example provided in (7), two tokens of ungrammatical *-le* in negation with *mei*, one token of ungrammatical *-le* with indirect speech verbs, and one with a non-telic event.

(6)

| Wo | qunian | chang chi | *(le) | henduo | pizza |
|---|---|---|---|---|---|
| I | last.year | often eat | LE | many | pizza |

"Last year, I used to eat much pizza."

(7)

| Wo | zhu zai | Tower C. Wo | ziji | zhu | *(le) |
|---|---|---|---|---|---|
| I | live at | Tower C. I | self | live | LE |

"I lived in Tower C. I lived by myself."

Although a variety of habitual activity verbs (e.g., *paobu* "run", *youyong* "swim", *wan* "play") and state verbs (e.g., *juede* "feel", *zhidao* "know", *you* "have") were provided as prompts from the role-play tasks, all oversupply cases with habitual and state verbs pertained to *chi* "eat" and *zhu* "live".

There were nine tokens of grammatical *-le* use, two with RVCs, six in the V-*le*-NM environment, and one with a specific, telic event. While there were few tokens of grammatical *-le* used in task, this is within expectation, since the role-play sheets were designed to help learners induce rules of *-le* forbidden context, and *-le* obligatory contexts were provided for the purpose of comparison. Additionally, there were two tokens of non-target-like *le* usage in V-*le*-NM environment, with errors due to word order. An example of word order error is given below as (8). Arguably, despite the word order errors, participants still showed awareness in the V-*le*-NM structure in these cases.

(8)

| Wo zai | pida | zhu le | zhu | le | zai sushe | liang | nian | le |
|---|---|---|---|---|---|---|---|---|
| I | at University.of.Pittsburgh | live LE | live | LE | dormitory | two | year | LE |

"I have lived in the University of Pittsburgh's dormitory for two years."

Finally, for the written editing task, Group C participants' scores and accuracy rate in the *-le* obligatory and *-le* forbidden contexts were examined. In the *-le* obligatory sentences, participants' mean score was 8.83 out of the maximum score of 12, with an accuracy rate of 73.6%. In the *-le* forbidden sentences, the mean was 6.92, with an accuracy rate of 57.6%. One sample, one-tailed *t*-test in comparison with a chance level of 0.50 suggests that the participants' performance in *-le* obligatory context was higher than the chance level ($t(11) = 7.34$, $p < 0.0001$), whereas their performance in *-le*

forbidden context was no more than chance (*t*(11) = 1.35, *p* = 0.102). In other words, there was no evidence that Group C participants had any awareness regarding constraints of *-le* usage.

In sum, beginning level learners' non-target-like performance in *-le* usage manifested in different ways in different tasks. In judgment tasks, their difficulties lay primarily in identifying and correcting errors of perfective *-le* in both obligatory and forbidden contexts. In the role-play task, underuses occurred more often than overuses, and occurred in the context of different verbs and eventualities (quantified, resultative, and specific event). Meanwhile, *-le* overuses appeared to be associated more clearly with certain verbs than others. Other *-le* usage errors related to word order were rare. Finally, results from Group C's written editing task indicate that participants, without experiencing focused treatment, lacked explicit knowledge regarding forbidden contexts of *-le*.

### 4.2. Effect of Consciousness-raising

To assess if consciousness-raising techniques (including explicit marking of verb-aspect forms and rule induction) led to different results than unfocused interactive activities of role-play, we first examined Group E participants' scores and accuracy rate in the written editing task. Participants' mean scores in *-le* obligatory context was 8.38 out of a maximum score of 12, with an accuracy rate of 69.9%. Participants' mean scores in *-le* forbidden context was 8.92 out of a maximum of 12, with an accuracy rate of 74.4%. One sample, one-tailed *t*-test was performed on accuracy rate in both contexts to make comparisons with the 0.50 chance level. Results showed that participants performed higher than chance level in both contexts: (*-le* obligatory context: *t*(12) = 3.94, *p* = 0.001; *-le* in forbidden context: *t*(12) = 4.46, *p* = 0.0004). In other words, Group E participants showed awareness regarding when *-le* should or should not be supplied in this posttest task.

Next, Group C and Group E participants' performance in the posttest in each of the two situations (*-le* forbidden and *-le* obligatory) was compared. A two-sample *t*-test revealed significant differences between groups in *-le* forbidden context, *t*(23) = 2.13, *p* = 0.044, Hedge's *g* = 0.85. On the other hand, the two-sample *t*-test comparing the two groups' performances in the *-le* obligatory environment revealed no significant differences between groups, *t*(23) = 0.63, *p* = 0.54.

Table 4 presents a summary of participants' mean scores and accuracy rates in the two divergent *-le* contexts. Specifically, Group E participants had better performance than Group C in *-le* forbidden context, demonstrating a development of explicit knowledge regarding *-le* constraints through consciousness-raising.

**Table 4.** Summary of two groups' performance in posttest.

|  | *-le* Obligatory Context | | *-le* Forbidden Context | |
|---|---|---|---|---|
|  | Scores (standard deviations) | Accuracy rate | Scores (standard deviations) | Accuracy rate |
| **Group C** | 8.83 (1.34) | 73.6% | 6.92(2.35) | 57.6% |
| **Group E** | 8.38 (2.18) | 69.9% | 8.92 (2.36) | 74.4% |

The above shows that in comparison with unfocused role-play tasks, consciousness-raising facilitated participants' explicit learning regarding rules of *-le* forbidden context. Consciousness-raising did not lead to higher outcome compared to unfocused role-play in *-le* obligatory context. This is not surprising, given that participants engaged in rule induction of *-le* forbidden context only in this study. As participants had previously learned about specific rules of obligatory *-le* context, the inclusion of obligatory *-le* in the role-play material did not lead to the development of new knowledge. However, it could have facilitated rule induction by offering participants examples of comparison, as discussed in the next section.

A closer look also revealed that Group E outperformed Group C in all three types of *-le* forbidden environment in the posttest. Their mean scores (out of a maximum of 4) and accuracy rates are reported below in Table 5. Due to the small number of items in each of the three sub-contexts, no inferential

statistical tests were performed. This descriptive result is discussed in triangulation with findings for our third research question, namely the outcome and processes of participants' rule induction.

**Table 5.** Scores and accuracy rate in three *-le* forbidden situations in posttest.

| | Habitual Activities | | State Verbs | | Speech Verbs | |
|---|---|---|---|---|---|---|
| | Mean scores | Mean accuracy | Mean scores | Mean accuracy | Mean scores | Mean accuracy |
| Group C | 1.08 | 27.1% | 3.08 | 77.1% | 2.75 | 68.8% |
| Group E | 2.08 | 51.9% | 3.54 | 88.5% | 3.31 | 82.7% |

*4.3. Outcome and Processes of Interactive Rule Induction*

Our third question pertained to both the outcome and processes of participants' consciousness-raising rule induction. To examine "outcome", transcripts of participants' paired/small group discussions and their written responses were analyzed to see if they had successfully induced the rules of *-le* forbidden context in the three situations. Table 6 gives a summary of the rules induced by participants, corresponding to the actual linguistic rules. Small groups/pairs within Group E are referred to as "pair 1" to "pair 6" below, though one of the "pairs" involved three participants.

**Table 6.** Rules induced by participants, corresponding to target language rules [5].

| Target Language Rules: *-le* Cannot Be Used | Rules Induced by Learners |
|---|---|
| #1: with habitual activities | • We do not use *-le* when we talk about habit (Pair 1);<br>• Not used when you are just saying "I used to … " (Pair 6) |
| #2: with expressions of (in)direct speech ("say" verbs); Alternatively, with verbs that take a verb construction as their object | • Not used with *gaosu* ("tell"), *wen* ("ask"), *shuo* ("say") (Pair 1);<br>• (Not used with) "speaking words" or "descriptive objects" (Pair 2);<br>• Not used for previous statements (Pair 4) |
| #3: with state verbs | • Not used with passive verbs; *juede* ("feel") is like passive, *zhidao* ("know") is also passive (Pair 1);<br>• When something stays the same (or) stays constant, you do not use *-le* (Pair 2);<br>• (Not used) when the action cannot be completed, like "like" (*xihuan*) (Pair 3);<br>• If it is like an "opinion" you do not use *-le*. It is like "ongoing" (Pair 4);<br>• (Not used for) something that is felt, not physically done; focused on a state of being (Pair 6) |

Note. Quotes were directly taken from transcripts or participants' written response sheets, while words in parentheses were added for clarity.

All three target rules were induced by participants. This corresponds to findings in the posttest, where better a performance was observed in all three *-le* forbidden contexts among Group E in comparison with the Group C. Each pair, except for Pair 5, had some success in inducing the rules, though their verbalization of the rules sometimes remained at the level of *patterning* (summarizing specific form-meaning matching without being able to extrapolating it into formal rules), and was not always linguistically accurate.

---

[5] One pair of participants, Pair 4, also responded that -le should not be used in negation, with mei. While this observation adhered to examples in role-play, this was not considered successful rule induction, since participants had been introduced to that explicit rule in class before the study.

Next, participants' rule induction processes were analyzed. Participants' inputs were coded on two levels: (1) the types of cognitive processes involved (*noticing, categorizing, labeling, prior knowledge or experience, patterning, rule formulation, metacognition*) and (2) whether the process was successful (i.e., leading to rule formulation adhering to target language forms) or not (i.e., incorrect or misleading).

In many cases, the process of rule induction started with *noticing and reporting*. (9) and (10) provide such examples.

(9)
"Ta shuo," she asked him. Don't have *-le*. So how can I say this? (Pair 1)

(10)
They used it here "yesterday", after the verb. (Pair 3)

Participants' *labeling* can naturally lead to *categorization*. Below in (11), participants' induction processes generally followed a path of *noticing-> labeling->categorization-> contrasting*. (1A and 1B refer to the two participants in Pair 1. The same goes for references of other speakers in excerpts that follow.) In this study, *contrasting* is defined as one's effort to make comparisons between rules or features of two or more distinctive linguistic categories. As the focus of participants' rule induction was constraints of *-le* usage, and participants' prior knowledge involved obligatory use of *-le* only, *contrasting* was a useful strategy. In the example below, *categorizing* and *contrasting* can be considered as evidence of participants' achieving deeper awareness (Cerezo et al. 2016), as participants needed to make mental effort to activate several examples, make associations and connections, as well as make distinctions. At the end of the processes, participants arrived at a fairly insightful conclusion that "active verbs" but not "passive verbs" in the past time frame should take *-le*. Although this verbalization of the rule lacked sufficient linguistic accuracy, the participants were able to conceptualize the forbidden versus obligatory environment of *-le*, and developed a heightened sense of awareness through the process.

(11) 1A: It's [not] used when

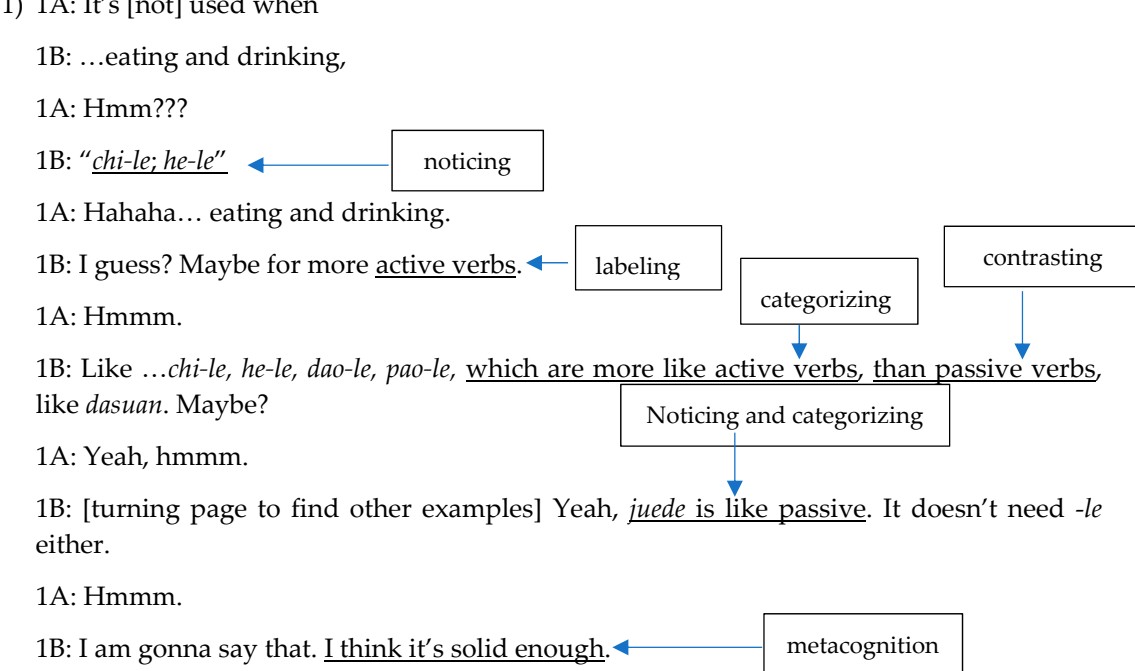

1B: …eating and drinking,

1A: Hmm???

1B: "*chi-le; he-le*"   ⟵ noticing

1A: Hahaha… eating and drinking.

1B: I guess? Maybe for more active verbs.   ⟵ labeling

1A: Hmmm.

1B: Like …*chi-le, he-le, dao-le, pao-le,* which are more like active verbs, than passive verbs, like *dasuan*. Maybe?   (categorizing / contrasting / Noticing and categorizing)

1A: Yeah, hmmm.

1B: [turning page to find other examples] Yeah, *juede* is like passive. It doesn't need *-le* either.

1A: Hmmm.

1B: I am gonna say that. I think it's solid enough.   ⟵ metacognition

Since *noticing* reflects low levels of awareness, and higher levels of awareness such as *categorization* require more amount of cognitive effort (Leow 2012), rule induction can progress linearly from *noticing* to other cognitive processes. However, the path of rule induction can also be interactive, especially in this study, in which rule induction was collaborative. For instance, below in (12), a participant hypothesized a pattern first, and then referred back to what they noticed in the role-play. This can

be seen as participant 4B's effort to integrate *noticing* with *patterning*, and it was also 4B's attempt to guide the partner to engage in *noticing*. In this excerpt, participant 4A made reciprocal contribution by offering his *prior experience* to support 4B's generalization. Both participants' input can be seen as acts of scaffolding or knowledge co-construction.

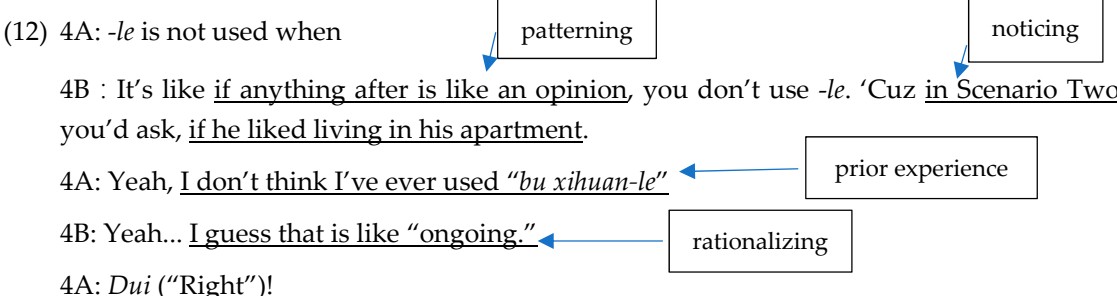

(12)  4A: *-le* is not used when

4B：It's like <u>if anything after is like an opinion</u>, you don't use *-le*. 'Cuz <u>in Scenario Two</u>, you'd ask, <u>if he liked living in his apartment</u>.

4A: Yeah, <u>I don't think I've ever used *"bu xihuan-le"*</u>

4B: Yeah... <u>I guess that is like "ongoing."</u>

4A: *Dui* ("Right")!

While the above examples show successful consciousness-raising through pair interaction, rule induction without instructor guidance can be unsuccessful. First of all, *noticing* did not always lead to successful rule formulation. In the following, unsuccessful rule formulation appeared to be an effect of negative L1 transfer. Both participants felt that it made sense if *-le* was forbidden in past time frame when a statement remained true, which is a linguistic pattern of the English past tense inflection (*-ed*).

(13)  4A: I don't know why it's not used in this.

4B: "Travel agency said," . . . *shuo* . . . , ah, probably because it's the listing.

4A: Oh, like the price doesn't change?

4B: It's still $800, even though they said it in the past.

4A: Oh, so like, the actual thing, like the thing they are saying still didn't change. 4B: Still true.

4A: Oh yeah, that makes sense.

In some cases, negotiation within pairs could be unhelpful. In excerpt (14), 2A initially started with a *labeling* that was descriptively close to the target rule, mentioning "speaking word". He also had a rough sense that the main verb takes a clausal object in those *-le* forbidden contexts.[6] However, while 2A struggled to conceptualize and verbalize linguistic features and categories, the label "descriptive object" was inaccurate, leading to misunderstandings among group members, and the attempt was unfruitful. (Only relevant part of the transcript is shown.)

(14)  2A: I feel like they are more "speaking" word.

2B: 'Cuz <u>you have *juede*, and *zhidao*.</u>

2C: <u>We can't categorize it.</u>

2B: I don't know.

[ . . . . . . ]

2A: [pause] They are objects that are described.

---

6    "Say" verb or "speaking word" itself does not immediately make *-le* forbidden in the environment. While Duff and Li (2002) referred to it generically as a "say" verb environment, it is the expression of (in)direction speech that would make *-le* generally unacceptable. For instance, "say" verbs taking a quantified noun phrase such as *wo shuo-le ta jiju* ('I scolded him a bit.') obligatorily takes *-le* because it fits the V-*le*-NM pattern and it does not express (in)direct speech. Similarly, it is arguable if verbs staking clausal object constitute *-le* forbidden environment. Despite the potential inadequacies of these attempted rule formulations, it is clear that participants were starting to uncover the linguistic features associated with *-le* and that their observations came close to generalizations made by some linguists and grammarians.

2A: [pause] It's like an adverb . . . ; it's like an adverb and a verb.

2B/2C: I don't know.

2A: This one's like describing "when." This one's describing like . . . .

2B/2C: A descriptive adjective. Descriptive adverb?

2C: I am making up new words.

2A: I don't know how to describe it.

2C: Wait. What did you say before?

2A: Oh, it was basically like, all of the . . . all of the like . . . actions are being described. So like like, the Beijing part is, like, for the summer. Like, it's like describing how long you are doing it. And then, for the travel agency, you are describing, like how much, like, the ticket cost. So like, You are describing . . . the action based on how long it takes or how much it costs.

2B: So, like, if you said, *wo chi-le yi-ge pingguo* versus *wo chi* . . . [ . . . . . . ]

2B/2C: *wo qu shangdian mai de* . . .

2A: *mai-wan*, so like . . .

2B: No, that doesn't make sense. <u>You can say *mai-wan-le*</u>

2A: Oh yeah.

2C: So you can say like "Descriptive object." "Descriptive objects" basically?

2A: Yeah, so it's like the object is being described.

Furthermore, due to the complex structures of the *-le* forbidden environment, participants also had difficulty categorizing different *-le* forbidden situations. Excerpt (15) below from Pair 3 can be considered "unsuccessful categorization". Participants attempted to consider different types of verbs (speech verbs, state verbs) as one category, making it difficult for them to identify actual patterns.

(15)  3A: I don't know. 'Cuz this one is like "I planned to go"; "he told me", which is completed, but they didn't use it when it's the past. But "he told me . . . "

3B: Yeah. They used it here yesterday, after the verb.

3A: Ah, you like something. Or you . . . , hmmm. I think it might be something like . . . it's an action, but it's like . . . .

3B: You can't complete it. Right? Something like that.

3A: (Yeah) . . . Like he told me. But . . .

3B: Yeah. like you cannot complete "told."

3B: So, not used when action . . . Just put like [you cannot use *-le* with verbs like] "like" and "asked"

In sum, participants' interactive rule induction led to some level of success in explicit rule formulation. Whereas rule induction processes can start with *noticing* and proceed to deeper levels of processing such as *categorizing* and *patterning*, the induction process within pairs/small groups was interactive in nature, with evidence of knowledge co-construction when the process was successful. Meanwhile, negotiation within pairs/small groups was not always fruitful.

## 5. Discussion

### 5.1. Learner Language Features

Learners' challenges in using *-le* manifested in different ways in different tasks in this study. First of all, results indicate that in beginner learners' oral production, *-le* underuse may be a more problematic area than overuse. While there were only 9 prompt verbs in the *-le* obligatory environment versus 16 prompt verbs in the *-le* forbidden environment in the role-play instruction, underuse was the most common error type. In the *-le* obligatory RVC environment, participants correctly supplied *-le* in only two tokens, missing *-le* in nine other tokens. The results corroborate findings from earlier studies that elicited oral production data. For instance, in Duff and Li (2002) film-based oral narrative task, native speakers provided plentiful RVCs, all marked with *-le*, whereas learners produced much fewer RVC tokens and missed *-le* 10 times while correctly providing *-le* only 8 times in the RVC environment. Wen (1995) also noted that beginner-level learners often missed *-le* in RVCs and hypothesized that learners may consider the resultative complement "as an indicator of completion", thus omitting *-le*.[7] Wen's (1995) hypothesis is probable, and in fact, learners would not be entirely wrong if they had such an analysis of resultative complement. According to Xiao and McEnery (2004), RVCs mark the "completive aspect", whereas *-le* marks the "actual aspect". In some situations, a resultative complement can replace *-le* because the two play the same functions in "perfectivis[ing] a situation" (p. 166). For instance, in (16a-b) and (17a-b), either the resultative complement or perfective *-le* can make the following sentences grammatical.

(16)

a.

| Wo | chi-wan | fan | jiu | qu kanshu |
| I | eat-finish | rice | then go | read book |

"I will go read books right after I finish my meal."

b.

| Wo chi | le | fan | jiu | qu kanshu |
| I eat | LE | rice | then go | read book |

"I will go read books right after I have had my meal."

(17)

a.

| Wo xiang-chulai | yi | ge | banfa |
| I think-out | one | CL | method |

"I thought of an idea."

b.

| Wo | xiang | le | yi ge | banfa |
| I | think | LE | one CL | method |

"I had an idea."

Given those cases where a resultative complement is interchangeable with *-le* in functions to bound an event or mark completion, it is not surprising that learners tend to drop *-le* in RVCs.[8] Compared

---

7    In these tokens of errors in Wen (1995), missing *-le* occurred in positions that were incidentally both verb-final and sentence-final. In many cases, either an aspect marker *-le* or sentence final *-le* is needed with RVC to indicate boundedness and completion, or change of state (e.g., *wo xuehui-le zhongwen; wo xuehui zhongwen le*, "I have learned and acquired Chinese").

8    In some RVC context, *-le* can be optional. For instance, *wo xiang-chulai (le) yige banfa* "I got an idea"; *ta dailai (le) yi-ben shu*, "He brought a book"; *wo zoujin wuzi* "I came into the room". The optionality can be explained if the "perfectivization" or boundedness function of *-le* can be achieved with certain complements such as these with [+punctual] features (Xiao and McEnery 2004). The optionality of *-le* in these RVC can make form-function mapping difficult for learners.

to other obligatory context of *-le* such as in V-*le*-NM or V1-*le*(O)-V2 structures, achieving native-like performance in using RVCs with *-le* may be especially difficult for learners.

The findings of more underuse than overuse in oral production but not necessarily in written editing also parallel with Duff and Li (2002). While the corpus study by Xu et al. (2019) referred to overuse as the most prevalent error type in learner speech, the differences may be due to the error tagging methodology used in corpus research.[9] Another possibility is that beginner-level learners in their very initial stage of acquisition tend to undersupply *-le*, whereas oversupply is an error type associated with more "intermediate"-level learners. In Yuan's (2019) study comparing her experimental and control groups' performances in posttest, she found that her experimental groups who had experienced learning tasks overused *-le* significantly more than the control group. Yuan (2019) explained that overuse can be considered learners' effort to experiment with the structure, a sign of language development, whereas underuse reflects an avoidance strategy and thus indicates an earlier developmental stage. As participants in this study had learned Chinese for approximately eight months, it is likely that they had not yet developed into the stage of experimenting with more usages. Future cross-sectional studies are needed to test learners' underuse and overuse in different developmental stages of acquisition.

Participants' oral production also indicates that *-le* oversupply may be associated with specific verbs. Specifically, participants oversupplied *-le* with *zhu* "live", but not with other state verbs such as *xihuan* "like". Participants also oversupplied the marker in *chi-le* in habitual activities but not with other verbs (*youyong* "swim"; *yundong* "physical exercise"). The specific semantic features of verbs may play a role here. *zhu* "live", for instance, is an "interval state" (physical or spatial configurations similar to *sit*, *stand*, etc.), in the words of Dowty (1979). It is different from mental state verbs such as "like" or "feel". In English, "live" can be in progressive aspect, and in both Chinese and English, *zhu* may be more prone to expressions such as "for a certain period of time" (thus requiring *-le* in Chinese) than mental state verbs. In Yang et al. (1999), all six tokens of state V-*le* tokens in their corpus were associated with *zhu* "live" and they occurred in V-*le*-NM structures such as *wo zai shanghai zhu-le liangnian* "I lived in Shanghai for two years". In other words, learners may have encountered forms of *zhu-le* in V-*le*-NM structure, while such usages of *xihuan-le*, *juede-le* are rare or ungrammatical. As to why *(changchang) chi-le* occurred in learner production but not *youyong-le* or *yundong-le*, one possibility could be their different levels of eventiveness: *chi* is a prototypical verb in Chinese, with high "eventiveness" (Monahan and Brunson 2014), whereas "swim", "exercise" can be nouns indicating events in Chinese. As *-le* is a "dynamic" particle (Liu et al. 2008, p. 137), learners may have more tendency in using it with verbs that are more prototypically associated with dynamic features. Another possible explanation could be participants' L2 exposure: in the textbooks that participants used, *yundong* was initially introduced as a noun. When it was used as a verb, it occurred in a habitual activity context (*yige xingqi yundong liang san ci*, "exercise a couple of times a week"). Similarly, *youyong* was initially introduced in a nominal-like way in the textbook (e.g., *ni qu youyong ba*, "How about you do swimming as an exercise?"). In addition, it is reasonable to expect that *yundong* and *youyong* occurred in much lower frequencies with *-le* than *chi-le* in L2 learners' exposure. L2 participants in this study may perceive *youyong* and *yundong* without *-le* as the more salient usages. If so, then the absence of *-le* oversupply with these verbs did not indicate acquisition; rather, participants likely had not developed an awareness regarding usages or prohibitions of *-le* yet, and were simply not using *youyong/yundong* with *-le* because it was not a prominent usage in their input.[10] In sum, oversupply and absence of *-le* may be related to

---

9    Xu et al. (2019) relied on the corpus' existing tags to code errors and found their data by extracting all samples containing *-le*. They found zero tokens of *-le* underuse in their corpus of 443,712 tokens. This is surprising, as the researchers themselves acknowledged, and they suggested further investigations in future research.

10   I thank an anonymous reviewer for pointing out this plausible explanation from the L2 input perspective. It is also interesting to note that among the six grammatical prompts in the grammaticality judgment pretest, participants had the lowest performance in item #4 (*wo zuotian da-le lanqiu, hai you-le-yong*, "yesterday I played basketball and swam"), with 11 out of the 25 participants (44%) correctly judging it to be grammatical. In comparison, average accuracy rate for the six grammatical

finely categorized semantic features of specific verbs or prominent usages in participants' L2 exposure. These potential explanations warrant further investigations in future research.

Overuse and underuse are not competing categories of errors, and pedagogical attention is needed for both. The written editing task results show that beginner-level learners had little to no knowledge regarding when *-le* should not be used, resulting in only chance level performance in *-le* forbidden sentences in the task. Explicit learning of *-le* constraints can help learners differentiate *-le* from past tense marker and encourage more target-like uses of perfective *-le*.

## 5.2. Consciousness-Raising

To facilitate the learning of explicit knowledge, consciousness-raising in this study was implemented through both explicit markings on the role-play sheet, and rule induction sessions within pairs/small groups. Compared to earlier studies (Wagner and Toth 2013; Yuan 2012), the present research is unique in its attempt to maximize student agency and autonomy, and there could be several advantages. First of all, without teacher mediation, group members had equal status as non-experts and they may be more at ease in contributing to the rule induction process. Secondly, compared to guided rule induction, where learners engage in form-function mapping tasks with examples and categories readily available (e.g., Wagner and Toth 2013; Yuan 2012), the tasks involved in unguided rule induction were more challenging. In finding relevant examples from role-play, coming up with appropriate categorization, participants needed to stretch the limits of their prior knowledge. After interpreting language samples, they also needed to convey meaning to others. In verbalizing their "analyses" and offering explanations to peers, they needed to construct clear and coherent representations of their own understanding (Van Lier 1996). This in itself required participants to build on higher levels of awareness. Third, in paired/small group rule induction, there was evidence of scaffolding within pairs and small groups through peer interaction, and they helped fill in each other's knowledge gaps, as we witnessed in some examples. Participants' cognitive process appeared multi-directional in this study, and *noticing* (or referencing to what one had noticed) was intertwined with one's rule or hypothesis formulation (i.e., *patterning*) and *rationalizing*. While the effectiveness of rule induction is generally believed to be relevant to greater depth of processing (i.e., one's increased mental effort in problem-solving) (Craik and Lockhart 1972), with the interactive and challenging natures of small group analytical talks, unguided rule induction can lead to particularly meaningful and memorable experiences that helps learning.

The use of small group/paired rule induction without instructor mediation had obvious limitations. As Table 6 illustrates, participants' rule formulation may not have always been descriptively accurate, even if they had developed some insight into the underlying mechanisms governing linguistic phenomena. As learners' competency in "languaging" varied, they may also have used informal or inaccurate terminologies due to incomplete metalinguistic knowledge, and this may lead to misunderstandings or ineffective negotiation, as shown by excerpt (14). Learners are not always attuned to peers' needs and it is possible that some group/pair interactions may be dominated by one particular member. For instance, in the successful induction example of (11), it remains unclear if speaker B's rule formulation was explained in accessible ways to speaker A. Without expert guidance, there was a lack of opportunities to check if group members had understood the rules inducted. Further, the effectiveness of small group/pair rule induction may be dependent on specific group dynamics. In this study, one particular pair did not succeed in inducing any of the target rules. A review of their interaction transcript indicated that the pair was not fully committed to the task and relied primarily on prior knowledge instead of observations and analytical talks. Nevertheless, as participants in the

---

prompts was 74%. In other words, participants might be generally unfamiliar with the *youyong* in the perfective form, and were thus unlikely to oversupply *-le* with this verb in prohibitory environment in the role-play task.

consciousness-raising condition had better performance in the posttest than the control group, the study confirmed the effectiveness of student-centered interactive rule-induction within small groups.

## 6. Limitations and Implications

The present research studied beginner-level learners' usage and errors of *-le* in interactive oral tasks and in written editing tasks. We found that consciousness-raising through explicit form making and student-centered rule induction supported explicit learning of *-le* constraints. The study was carried out in a real classroom setting, and the pedagogical intervention was short and effective. As Yuan (2012, p. 85) pointed out, consciousness-raising sessions that stretch over several class periods may not represent actual practices that teachers can implement. Thus, short pedagogical intervention sessions have high ecological validity. Though the number of stimuli used in each task in this study were small, they included rules of aspect marker *-le* in a variety of obligatory and forbidden contexts, so that results can meaningfully indicate learner language patterns. Future studies on learners' *-le* acquisition can further investigate learners' use of the aspect marker with RVCs and with specific verbs, and include considerations of the *-le* optional context. Due to the timing of the experiment, a delayed posttest was not implemented, but future researchers should attend to this issue, so as to assess the extent to which learners internalize explicit knowledge gained through consciousness-raising. This study is one of the first few to use an unguided interactive rule induction approach, and the comparative effect of guided instruction versus student-centered or self-guided rule induction should be further studied.

Development of linguistic competence in complex grammatical rules such as aspect *-le* is a slow process, requiring elaborate and intensive treatment. The techniques used in this study can provide a successful example of embedding focus-on-forms in language tasks in the classroom. It is also hoped that this classroom-based research takes one step further in the field's effort to bridge the gap between controlled lab-based experiment and natural interactions in language classrooms.

**Funding:** This research received no external funding.

**Acknowledgments:** The author wishes to thank the participants of this study for their time. She is also immensely grateful for the three anonymous reviewers and the guest editor for their constructive feedback. All errors are her own.

**Conflicts of Interest:** The author declares no conflict of interest.

## Appendix A. Grammaticality Judgement Pretest

The original pretest was given in Chinese characters, with *pinyin* on top. Ungrammaticality is indicated by * here and not in the task. English translations for stimuli are provided here and not in the actual task. Filler sentences are not included below.

*Instructions*: Judge the grammaticality of the following sentences. If it is grammatical, put "C" for "correct" in the parentheses. If the sentence has a grammatical error, put "I" for "incorrect". Correct the errors for those incorrect sentences by editing directly on the sentences.

(C)    1 *Wo xiaoshihou hen xihuan chongwu, danshi xianzai wo mei yang chongwu.* "I used to like pets when I was little, but I do not have pets now."

(I)    2 *Zai jiazhou shixi de shihou, wo youde shihou qu (*le) zhongcanguan chifan.* "When I was in California for my internship, I sometimes went to Chinese restaurants to eat."

(C)    3 *Xiao Ming gaosu wo jintian you kaoshi, suoyi wo zhunbei de henhao.* "Xiao Ming told me that there is a test today, so I prepared very well."

(C)    4 *Wo zuotian da le lanqiu, hai you le yong.* "Yesterday I played basketball and swam."

(I)    5 *Zhongxue de shihou wo xiwang (*le) xue xibanya yu, keshi xianzai wo xue zhongwen.* "When I was in middle school, I hoped to learn Spanish, but now I study Chinese."

(C)    6 *Wo zhongwu he le san-bei kele, xianzai bu ke.* "I drank three cups of coke at noon, and I am not thirsty now."

(I)     7 *Xiao Gao shuo (\*le) ta bu qu jintian de wanhui, yinwei ta mei shijian.* "Little Gao said that he would not go to tonight's party, because he does not have time."

(I)     8 *Wo yijing zuo-hao \*(le) jintian de zhongwen gongke.* "I already finished today's Chinese homework."

(C)     9 *Wo qunian xuexi hen mang, suoyi changchang 12 dian cai shuijiao.* "I studied very busily last year, so I often went to sleep as late as 12AM."

(I)     10 *Zuotian wanshang wo he pengyou yiqi kan \*(le) "Harry Potter."* "My friend and I watched Harry Potter together last night."

(C)     11 *Zhe pian kewen youdian'er nan, ni kandong le ma?* "This text is a bit difficult; do you understand it (after reading)?"

(I)     12 *Zuotian de wanhui, Xiao Zhang chi \*(le) shi ge jiaozi.* "Xiao Zhang ate ten dumplings at yesterday's party."

## Appendix B. Instructions and Sample Scenarios in the Role-play Sheet for Group E

In the role-play sheets participants received, Chinese characters instead of *pinyin* were shown for each prompt verb. Instructions for two scenarios were provided as examples, while the original role-play sheet contained four scenarios.

*Instructions*: The following are scenarios that you would act out with your partner, with one person taking up the role of either A or B. Use Chinese only to act out the scenario.

If the key verbs in parentheses appears as (VØ) such as (*zuo Ø*), it means *le* is not used. If it appears as (V-*le*) such as (*zuo-le*), *le* should be used. Pay attention to these when role-playing.

After completing all the four scenarios, turn to page 3 and discuss with your partner regarding the use of –*le*.

*Appendix B.1. Scenario One*

A: Last week, you and your friends made a number of plans from Monday to Thursday, including going shopping, playing tennis, going to a movie, having dinner, etc. But one of your friends did not show up in any of the events. Find out why he/she missed for each of these activities. (*weishenme mei* **VØ**)

B: You previously made plans with your friends for several activities last week, but didn't go to any, because of the following on different occasions:

You felt unwell (*juede Ø*);

You knew that the weather was going to be bad (*zhidao Ø*);

You took the wrong bus (*zuocuo **le** che*);

You ruined your stomach by having bad food (*chi-huai **le** duzi*).

Apologize and propose new plans with your friend.

*Appendix B.2. Scenario Two*

A: You are new to campus living. Talk to your friend, who lived in a dorm in the past but just moved out. Ask him/her about his/her former dorm life, including:

where he lived (*zhu Ø*)'

how many roommates he had (*you Ø*) and if he/she frequently spent time with the roommate(s) (*gen . . . yiqi wan Ø*)'

If he frequently ate out (*zai waimian chifan Ø*);

If he liked (*xihuan Ø*) living in an apartment;

Further, ask him/her why he moved.

B: You lived in a dorm last semester but moved to an apartment this semester. Your friend asked you about your former dorm life. Use *zhu*; *you*; *wan*; *chifan*; *xihuan*, etc. to answer his/her questions.

Then, tell him that you decided to change to an apartment now because you had been at Pitt for two years (*xuexi **le***) and had lived in a dorm for two years (*zhu **le***), and you want to live in someplace new.

**Appendix C. Written Editing Posttest**

The original posttest was given with Chinese characters and pinyin on top. [+le] indicates the 12 obligatory contexts for -*le* usage. [h], [s], and [i] respectively indicate cases where -*le* is ungrammatical due to being in "habitual activity," "state verb," and "indirect/direct speech verb" contexts.

*Instructions*: Please provide *le* in the blank where it is needed. If *le* should be absent in that sentence, write /.

Qunian wo de nanpengyou Xiao Lin zai Zhongguo gongzuo [h], suoyi wo qu Zhonguo lvxing [+le] yi-nian. Wo xingqi-liu dao [+le] Beijing jichang yihou, juede [s] youdian ke. Kandao jichang li you [s] yi-ge Xingbake (Starbucks), jiu zou [+le] jinqu. Wo yiqian zai Meiguo hen ai [s] he kafei. Danshi wo bu xihuan [s] na bei kafei, suoyi wo huan [+le] yi-bei cha. Yibian he yibian deng Xiao Lin. Wo zai Xingbake zuo [+le] ban ge xiaoshi, Xiao Lin jiu lai [+le]. Xiao Lin wen [i] wo: "Ni zai feiji shang zuo [+le] shenme?" Wo shuo [i], "Wo zai feiji shang shui [+le] wu-ge xiaoshi de jiao, kanwan [+le] yi-ben shu." Xiao Lin gaosu [i] wo, ta you e you ke. Wo mashang shuo [i]: "Na women yiqi qu chi wanfan ba!" Na tian women dian [+le] henduo cai, zuihou ba qian dou yongwan [+le].

Zai Zhongguo de shihou, wo mei ge xingqi dou gei babamama da [h] dianhua. Wo ye kaishi xihuan-shang henduo xin dongxi. Biru, wo yiqian he [h] kafei, danshi xianzai ai he cha. Wo yiqian ye changchang chi [h] Faguo cai, keshi xianzai geng xihuan chi Zhongguo jiaozi. Wo hai xuehui [+le] xie hanzi.

*English Translations of the Passage (Not Provided in Posttest to Participants)*

Last year, my boyfriend, Xiao Lin, went to China to work, so I travelled and went to China for a year. I arrived at the Beijing Airport on a Saturday and felt thirsty when I arrived. I saw a Starbucks coffeeshop in the airport, and went in. I used to love drinking coffee when I was in America, but I did not like that cup of coffee, so I exchanged it for a cup of tea. I was waiting for Xiao Lin and drinking tea at the same time. I sat in Starbucks for half an hour before Xiao Lin came. Xiao Lin asked me, "What did you do on the plane?" I said, "I had a five-hour sleep on the airplane and finished reading a book." Xiao Lin told me that he was both thirsty and hungry. I immediately said that we (should) go get dinner. On that day, we ordered several dishes, and in the end spent all the money.

While in China, I called my parents every week. I started to like many new things. For instance, I used to drink coffee, but I now love drinking tea. I used to eat French food frequently, but now I prefer Chinese dumplings. I also learned how to write Chinese characters.

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
