# Peer review of "Perfective -le Use and Consciousness-Raising among Beginner-Level Chinese Learners"

_languages, doi:10.3390/languages5020016_

Round 1

Reviewer 1 Report

The study connects L2 Studies of -le in experimental environment and real teaching practice. It firstly investigates what errors tend to occur in beginner level CFL learners’ use of -le in their oral and written performances in language classrooms. Secondly, it explores the role of consciousness raising by using role-play sheets with explicit markings of language forms and by having learners engage in small group rule induction. I think it is generally well-written. However, there are also some concerns:

Forbidden context of -le:

“Say” verbs:   Wo gaosu (*le) ta, wo jintian bu qu xuexiao.

Under the context that someone asks whether you have told him that you would not go to school today (Ni you mei you gaosu ta……?). He can say Wo gaosu le ta, wo jintian bu qu xuexiao. It sounds a correct way of using -le here.

Dasuan (plan) also sounds like a state verb.

In section 2.1, lines 53 and 54, the author states that “-le also tends not to go well with ‘say’ type of verbs indicating direct or indirect speech, or verbs taking a clausal object (Duff and Li 2002, p.424-425)”. However, the sample sentence regarding to “say” verbs in Table 1 seems requiring more discussion. To me, -le can go with some “say” types of verbs, such as the examples below:

ta shuo le mingtian bu qu xuexiao.
他说明天不去学校。
ta dasheng han le ji ju.
他大声喊几句。

According to Duff and Li (2002), -le cannot be used after “expressions of direct or indirect speech, referred to generically as ‘say’ verbs”. For example:

Ta shuo-X ta mei qu-GUO Zhongguo
他说(-X)他没去过中国
[he say-X he NEG go-GUO China]
'He said he hadn't been to China',

In the example above, "-le" cannot be used probably because "mei qu-GUO" appears in the clause, not because of the “say” verb shuo. If the clause is:

ta shuo le ta bu xiang qu Zhongguo
他说他不想去中国。

The sentence is grammatically correct. The difference between ta shuo le ta bu xiang qu Zhongguo and ta shuo ta bu xiang qu Zhongguo lies in that the former is used to answer a question like whether he/she said this or not and the latter is just a simple statement.

Moreover, in table 1 the “le” forbidden environments include “verbs taking a clausal object”. This is arguable as there exists many exceptions where -le can go together with” verb + a clausal object”. For example,

wo tongyi le mingtian qu canjia bisai
我同意了明天去参加比赛。
ta daying le xi age xingqi qu shangke
他答应了下个星期去上课。

Both sentences are grammatically correct.

In table 1, the example Wo benlai dasuan (*le) mingnian qu Zhongguo, xianzai bu neng qu le states that -le is not necessary, because there is a clausal object after "dasuan打算". However, it may be because "dasuan打算" is a state verb, just like “觉得、认为、希望”. As mentioned by Lv and others, -le should not be used when the verb does not convey a state of change, such as “是、姓名、好像、属于、觉得、认为、希望、需要、认为……” (《xian dai han yu ba bai ci现代汉语八百词》, p. 352).

Author Response

  1. I thank the reviewer for his/her insightful observations. He/She pointed out some of sentences that were presumed to be in the forbidden context of -le do not appear to be exactly ungrammatical to him/her. Those are related to two constraints mentioned in Table 1. One is with “say” type of verbs expressing direct or indirect speeches verbs taking clausal objects.” In revision, I have revised Table 1 (and elsewhere) by referring to this “forbidden” environment as “expressions of direct/indirect speech” instead of “‘say’ verbs.” In expressing direct speeches, -le is generally forbidden in Chinese, and there tends to be no difference between direct and indirect speech in Chinese, except for the shift of personal pronouns.

Ta dui laoshi shuo: “wo shi di yi ge dao de.”

Ta dui laoshi shuo ta shi shi dian dao de.

(Examples from Hagenaar, 1996, p.293).

Hagenaar, E. (1996). Free indirect speech in Chinese. In Reported Speech: Forms and Functions of the Verb. Edited by T. Janssen and W. van der Wurff. Amsterdam: John Benjamins, pp.289-298.

Note that -le is not used in either direct or indirect speech above.

I agree with the reviewer that some exceptions can be found. An example given by the reviewer was “ta shuo le mingtian bu qu xuexiao” (‘He said he would not go to school tomorrow.’).

The sentence is acceptable in context to correct a wrong assumption, and with a pause after “说了”:

A: 我明天跟小明见面时会跟他聊聊。

B: 可是小明说了,他明天不去学校。

The manuscript contains an example in line 59.

Wo gaosu (*le) ta, wo jintian bu qu xuexiao.  (‘I told him that I would not go to campus today.’)

The reviewer felt that “le” may be permissible if it is used as a response to a yes/no question of Ni you mei you gaosu ta……?

To me, a more licit positive response to “Ni you mei you gaosu ta……?” is “Wo gaosu ta le” (with sentence final le) instead “Wo gaosu le ta …”.

Wo gaosu (*le) ta, wo jintian bu qu xuexiao” in general contexts is quite unacceptable, and the sentence immediately becomes grammatical if “le” is not present. That seems to indicate the constraint of not applying -le with expressions of direct/indirect speech is present, though exceptions in marked situations are possible. A quick corpus search using the Center for Chinese Linguistics PKU corpus also indicates that there are very few (though not non-existent) sentences with “说了 + indirect/direct speech,” while there are far more “说 + indirect/direct speech” examples.

For revision, I made a note (footnote 2) acknowledging that some exceptions in marked situations are possible. As experiment materials (including pretest, posttest, and scenarios in role play) do not involve marked situations that may license -le insertion expressing (in)direct speech, these exceptions should not affect the study’s general conclusion. 

  1. In my original submission in Table 1, I listed a fourth “forbidden environment” of -le with clausal object. That is according to Duff & Li (2002) and T’ung and Pollard (1982). I agree with the reviewer that this purported rule of “clausal object” may need reconsideration. In revision, in line 63-70, I introduced this “rule” as a generalization that remains arguable.

“Clausal object” constraint was not included as a “-le forbidden environment” in the experiment materials, and was not a “target rule” which participants were asked to induce. In revision, I took out reference to this purported rule in Table 1.

The “clausal object” constraint is still relevant to the present study. That is because some participants’ rule induction process came close to making observations similar to this purported rule claimed by T’ung & Pollard (1982) and Duff & Li (2002). The present study only intends to show that L2 language learners can develop explicit awareness regarding -le usage patterns through noticing and rule induction. In the relevant rule induction episode (line 512), I added a footnote, explaining that while these L2 participants appeared to have develop some insight, the “rules” formulated should most certainly be subject to modification.

I thank the reviewer for these comments. After these revisions, the claims made in the present study should be more solid.  

Reviewer 2 Report

 The acquisition of Perfective -le  is a very important topic for CFL learners, especially for beginning-level students. It is crucial to investigate patterns in the use of -le in authentic classroom tasks by beginner-level CFL learners within the framework of explicit learning and consciousness raising. I believe the methods and framework adopted in this study may serve as an example for examining CFL learners' acquisition of other grammatical constructions in Mandarin Chinese. I thoroughly enjoyed reading the paper. It is a very well-designed study and a very well-written article. Congratulations!

Author Response

Thank the reviewer for encouraging comments. 

Reviewer 3 Report

The topic is interesting and useful , and the study can provide practical pedagogical guidance. The argument and discussion are well presented. A few questions:

  1. typo: line 707, should be zhongwu instead of zhongwen
  2. The article argues that unguided small group rule induction supported participants’ learning of –le usage constraints, but also points out that participants’ rule formulation was not always accurate. Would the misunderstanding of the rules negatively impact the acquisition? Since the experiment group underwent consciousness raising through both explicit markings on the role-play sheet, and rule formulation discussion, would it be possible that the learners benefited from the explicit markings most, and not the rule induction?
  3. line 617: The article points out that chi-le occurred but not youyong-le or yundong­-le in the data. Other than the possibility of semantic features of specific verbs as mentioned, is it possible that it is because of the language exposure? In the textbook the students use, youyong and yundong were introduced in a habitual context, and they are not as high frequency as chi. Students thus perceive youyong and yundong without le as the more salient use. This would suggest students have not acquired the usage/prohibition of le analytically. It would be interesting to know the students’ performance on the Grammaticality Judgement Pretest #4, since it involves the usage of youyong with –le.

Author Response

Thank the reviewer for these valuable comments. I corrected the typo in line 707.

For the reviewer’s 2nd comment, both explicit marking and rule induction are mechanisms to implement consciousness raising, and the study by design combines the two mechanisms: without explicit marking, rule induction would be impossible. Admittedly, there is a difference between “effect of consciousness raising” and “outcome of rule induction.” The former is addressed in research question two (line 160 -161), and the latter is addressed in research question three (line 162). For the effect of consciousness raising, we do not separate the benefit of explicit marking and rule induction, and it was found to be beneficial (through comparisons of experimental group versus control group’s performance in posttest). For the outcome of “rule induction” (research question three), it is true that participants’ rule formulation is not always accurate. As the reviewer pointed out, it is indeed worth investigating in a future study if misunderstanding of rules may negatively affect acquisition. In this study, results (from comparisons of performance of the experiment and control groups) suggest that overall benefits were maintained, despite the fact that L2 participants rule formulation through induction was not perfect.

In his/her comment #3, the reviewer suggested that the absence of -le oversupply with youyong and yundong (‘swim’; ‘exercise’) may be relevant to how these words were introduced and used in participants’ textbooks. This is indeed a probable explanation. I added that as a possible explanation on line 630-642 in the revised manuscript. The reviewer also pointed out that participants’ performance in the grammaticality judgment task specifically in item #4 may be relevant. The reviewer was correct, and participants’ performance in that item appeared to lend support for the L2 input based hypothesis. I included the information in a footnote and thank the reviewer for that comment (footnote 10).